# Dual roles of the sterol recognition region in Hedgehog protein modification

Rahul Purohit [1], Daniel S. Peng [1], Erika Vielmas [1] & Alison E. Ondrus[1✉]

Nature provides a number of mechanisms to encode dynamic information in biomolecules. In metazoans, there exist rare chemical modifications that occur in entirely unique regimes. One such example occurs in the Hedgehog (Hh) morphogens, proteins singular across all domains of life for the nature of their covalent ligation to cholesterol. The isoform- and context-specific efficiency of this ligation profoundly impacts the activity of Hh morphogens and represents an unexplored facet of Hh ligand-dependent cancers. To elucidate the chemical mechanism of this modification, we have defined roles of the uncharacterized sterol recognition region (SRR) in Hh proteins. We use a combination of sequence conservation, directed mutagenesis, and biochemical assays to specify residues of the SRR participate in cellular and biochemical aspects of Hh cholesterolysis. Our investigations offer a functional portrait of this region, providing opportunities to identify parallel reactivity in nature and a template to design tools in chemical biology.

---

[1] Division of Chemistry and Chemical Engineering, California Institute of Technology, Pasadena, CA 91125, USA. ✉email: aondrus@caltech.edu

The Hedgehog (Hh) morphogens undergo an auto-catalytic modification that cleaves the translated protein approximately in half and covalently appends cholesterol to the last residue of the N-terminal fragment.[1] To date, this intramolecular small molecule transfer activity has been observed only in Hh proteins and in no other domains of life.[2] Cholesterol modification is fundamental to the activity of the Hh morphogen;[3,4] mutation of the catalytic residues are lethal, and mutations that reduce the efficiency of the reaction give rise to profound congenital disorders (Supplementary Fig. 1).[5,6] This unique modification appears to have co-evolved with cholesterol-sensing mechanisms that coordinate Hh signal transduction, chemically linking embryogenesis to cholesterol homeostasis.[7–9]

The initially translated, full-length Hh proteins (typically 350–475 residues in length) consist of an N-terminal signaling domain, an internal catalytic domain known as the Hedgehog/Intein (Hint) domain,[10] and a C-terminal sequence known as the Sterol Recognition Region (SRR, Fig. 1a).[11] During post-translational modification, the full-length protein is cleaved at the end of the N-terminal signaling domain; concomitantly, the last residue of the N-terminus is released as a cholesterol ester. Independently, the N-terminal residue of the Hh morphogen is palmitoylated by dedicated acyltransferase enzyme in the ER.[12,13] A crystal structure of the *D. melanogaster* Hint domain (residues 258–402) provided by Beachy and coworkers immediately revealed the basis for proteolytic activity of the protein: a splicing

unit remarkably similar in tertiary structure to inteins found in bacteria and fungi.[14,15] Hint domains catalyze intramolecular attack of a cysteine, serine, or threonine side chain on the backbone carbonyl of a preceding residue, effecting N–to–S/O acyl transfer to create a labile (thio)ester bond in the protein backbone. In protein-splicing inteins, a nucleophilic residue at the C-terminus of the protein reacts with the (thio)ester in a trans-esterification reaction, ultimately ligating the two exteins.[16,17] Non-splicing Hint domains lack residues required for canonical protein splicing, and instead undergo alternative mechanisms of (thio)ester hydrolysis or ligation. In Hh proteins alone, the thioester formed during the splicing reaction is intercepted by cholesterol, co-opting this mechanism for small molecule adduction.

As in many other Hint-containing proteins, the N-terminus of Hh does not participate in the cleavage reaction.[10] Thioester formation and cholesteroylation activities inhere solely in the Hint and SRR regions, respectively. Accordingly, cholesterol ligation can occur in vitro and in cells when the Hh N-terminus is replaced by fluorescent proteins or simply by a His-tag.[18–22] However, the molecular details of the SRR that facilitate cholesterol transfer remain unknown. The SRR, which spans 50–100 residues in different species, bears no clear homology to known proteins in any domain of life. While a recent computational study posits an intriguing connection between the SRR of *D. melanogaster* and the cryptogein protein of *P. cryptogea*,[23] this

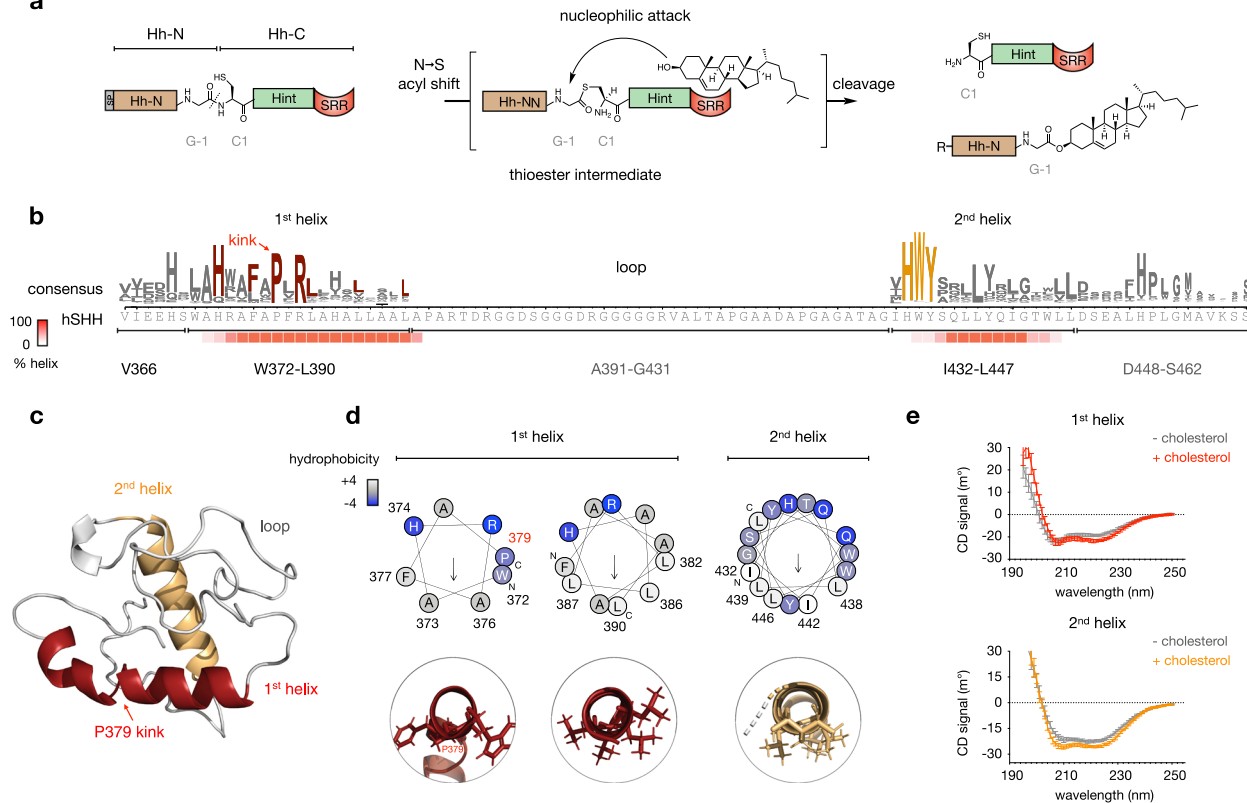

**Fig. 1 The SRR is a helix-loop-helix motif. a** The Hint domain within full-length Hedgehog (Hh) proteins catalyzes formation of a thioester intermediate in the peptide backbone. The SRR facilitates cholesterol attack on the thioester, cleaving the Hh protein and ligating cholesterol to the last residue of the N-terminus. In a separate process, palmitate is attached to the first residue of the N-terminus after cleavage of the signal peptide. Conventional Hint domain numbering is shown; R = palmitoyl. **b** Sequence logo for PROMALS3D alignment (ref. [24]) of the SRR of 700 manually annotated Hedgehog protein sequences above the corresponding sequence of hSHH (SRR, residues 363–462). Heatmap below shows the percent helical character predicted by the JNet4 algorithm (ref. [25]). **c** Model of the SRR of hSHH (residues 363–462), created using ab initio Rosetta prediction (ref. [26]). **d** Helical wheel diagrams of the 1st and 2nd SRR helices from HELIQUEST (ref. [26]). The 1st helix is twisted at P379; axial views of each segment are shown. Residues are colored according to the Kyte Doolittle hydrophobicity scale (white = hydrophobic, blue = hydrophilic). **e** Circular dichroism (CD) spectra of peptides encompassing the 1st helix (residues 368–391) and 2nd helix (residues 431–449) in liposomes.

model awaits analysis in a membrane environment and experimental evaluation. No high resolution structure of the SRR has been solved; moreover, no static conformation can fully capture the dynamic activity of these fascinating proteins.

To define structural requirements for cholesterol transfer by the Hh SRR, we used a combination of tools at the interface of chemistry and biology. We demonstrate that the process of Hh cholesterolysis encompasses two main activities: (1) engagement of cholesterol as a nucleophile, and (2) cellular localization to cholesterol-containing membranes. We use sequence conservation and structure prediction to reveal a helix-loop-helix arrangement that is broadly represented among Hh SRRs. Guided by this structural analysis, we used site-directed mutagenesis of these motifs within human Sonic hedgehog (hSHH) expressed in mammalian cells to identify loss of function mutations that abolish Hh autoprocessing. Biochemical analysis of full-length protein purified from mammalian cells enabled us to distinguish conserved hydrophobic residues that are essential for cholesterolysis in cells but not in vitro. Cellular localization of a fluorescent SRR fusion protein revealed a Golgi targeting motif that may bias the interaction of full-length hSHH with cellular membranes for cholesterol recruitment. By defining the structural requirements and elementary steps in this unconventional modification, this work expands our understanding of Nature's chemical biology toolkit and facilitates rational design of tools for small molecule ligation in cells.

## Results

**A helix-loop-helix motif is conserved across SRRs**. To identify the residues within the SRR that participate in cholesterol transfer we considered the SRR as V366-S462 of the hSHH protein, which begins after the last corresponding residue in the crystal structure of the *D. melanogaster* Hint domain.[10] Cleavage of the *D. melanogaster* protein at this residue renders the protein soluble in the absence of detergent and eliminates cholesterol ligation during proteolysis. As protein-based and nucleotide-based alignment programs show no discernible homology between the hSHH SRR and other non-Hh proteins, we aligned 700 manually curated, diverse Hh proteins to identify common features across species and isoforms (Fig. 1b and Supplementary Fig. 2).[24] This alignment revealed two sections of conserved residues before and after a long intervening sequence that is unique to a subset of Hh proteins (e.g., hSHH). Secondary structure analysis revealed that the conserved segments are predicted to form α-helices,[25] while the glycine-rich intervening segment is predicted to be disordered.

To guide our hypotheses regarding the residue-level contributions of the SRR to cholesterolysis, we used ab initio fragment assembly to generate a 3D structure of the last 100 residues of hSHH, which encompasses the hSHH SRR (Fig. 1c).[26] In line with our secondary structure predictions, a lowest energy hSHH SRR structure possesses helices at residues from W372-L390 (1st helix) and I432-L447 (2nd helix), which are linked by a disordered loop comprising residues A391-G431. The structure contains a "kink" introduced by proline residue P379 in the 1st helix, which separates the 1st helix into two angled segments from W372-A378 and P379-L390.[27] Helical wheel analysis of the two regions, revealed that each segment of the helices possess a face enriched in hydrophobic residues (Fig. 1d). This arrangement is characteristic of amphipathic peripheral membrane helices;[28] none of our analyses predict these sequences to span the bilayer.

To gain empirical support for our alignment-based model, we synthesized peptides comprising the 1st and 2nd helices and examined their secondary structures by Circular Dichroism (CD) spectroscopy (Fig. 1e). As anticipated, these hydrophobic peptides

are insoluble in the absence of liposomes. CD spectra of each helix in multidisperse POPC liposomes clearly displayed helical character in both the presence and absence of cholesterol. Grounded by conservation analysis and secondary structure analysis, we used this model as a scaffold for experimental studies to define residue-level structural elements of the SRR that link Hint-catalyzed proteolysis to cholesterol transfer.

**Cellular cholesterolysis relies on conserved SRR residues**. As a relatively hindered, hydrophobic nucleophile with a pKa of ~18,[29] the cholesterol hydroxyl group imposes strict requirements on the residues involved in nucleophilic activation and trajectory for attack. Whereas a Hint-SRR unit catalyzes cholesterolysis both in cells and in vitro, Hh mutants lacking the SRR undergo proteolysis without attachment of cholesterol.[10] These observations demonstrate that the SRR is necessary and sufficient for biochemical Hh cholesterolysis. In cells, however, the marked insolubility of cholesterol in aqueous solution (<10 nM)[30] and cellular mechanisms of cholesterol sequestration also require mobilization of cholesterol from cellular membranes.[31] Accordingly, we hypothesize that conserved helices within the SRR fulfill two functions: (1) facilitate biochemical attack by cholesterol, and (2) recruit the full-length protein to cholesterol-containing membranes in the cell.

To assess the propensity of mutant hSHH proteins to undergo cholesterolysis in mammalian cells, we overexpressed hSHH and mutants in HEK293T cells. In this cell type, which lacks accessory proteins involved in native secretion of cholesteroylated hSHH, differently processed and/or lipidated forms of the hSHH protein partition into cell lysates or secreted media. The cholesteroylated and dually lipidated (cholesteroylated and palmitoylated) N-terminal proteins remain membrane-associated and are retained in cell lysates. By contrast, hSHH bearing the palmitate modification alone is secreted into cell media.[11,32–35] Accordingly, we used production of cell-associated hSHH-N in lysates (hSHH-$N_L$) as a measure of cholesteroylation for a given SRR mutant (Supplementary Fig. 3a, b).[36]

We first generated deletion mutants that lacked components of the helix-loop-helix motif identified in our protein alignment and biochemical studies (Fig. 2a). Consistent with previous observations, a construct lacking the SRR failed to produce any hSHH-N in cell lysates.[10] Cell-associated hSHH-N was likewise absent in mutants lacking the 1st or 2nd helices. Strikingly, a construct that lacked a non-conserved 32-residue connector (Δloop, residues A393-P424) functioned identically to the wild-type protein. This result defines the 1st and 2nd helices as a minimal required unit, consistent with the variable size of the loop region in Hh proteins known to undergo Hh cholesterolysis.[37]

To more explicitly probe the residues of the SRR that participate in cholesterolysis in cells, we performed alanine scanning of both the 1st and 2nd helices (Fig. 2b, c). Mutation of highly conserved 1st helix residues H374, F377, and R381 drastically reduced production of cell-associated hSHH-N. Interestingly, mutation of the absolutely conserved P379 residue did not reduce the proportion of cell-associated hSHH-N relative to WT but lowered the total amount of protein in cell lysate, likely due to an effect on biochemical and/or cellular protein stability.

Based on our helical wheel analysis (Fig. 1d), we envisioned that the three conserved leucine residues on the same face of the 1st helix, L382, L386, and L390 might interact with the lipid tails of the membrane. Consistent with this hypothesis, Monte Carlo simulations of the interaction between the 1st helix peptide (W372-L390) and a phospholipid bilayer predict that these three leucines will reside at the membrane interior (Fig. 2d).[38] While single L382A, L386A, and L390A mutants enabled the

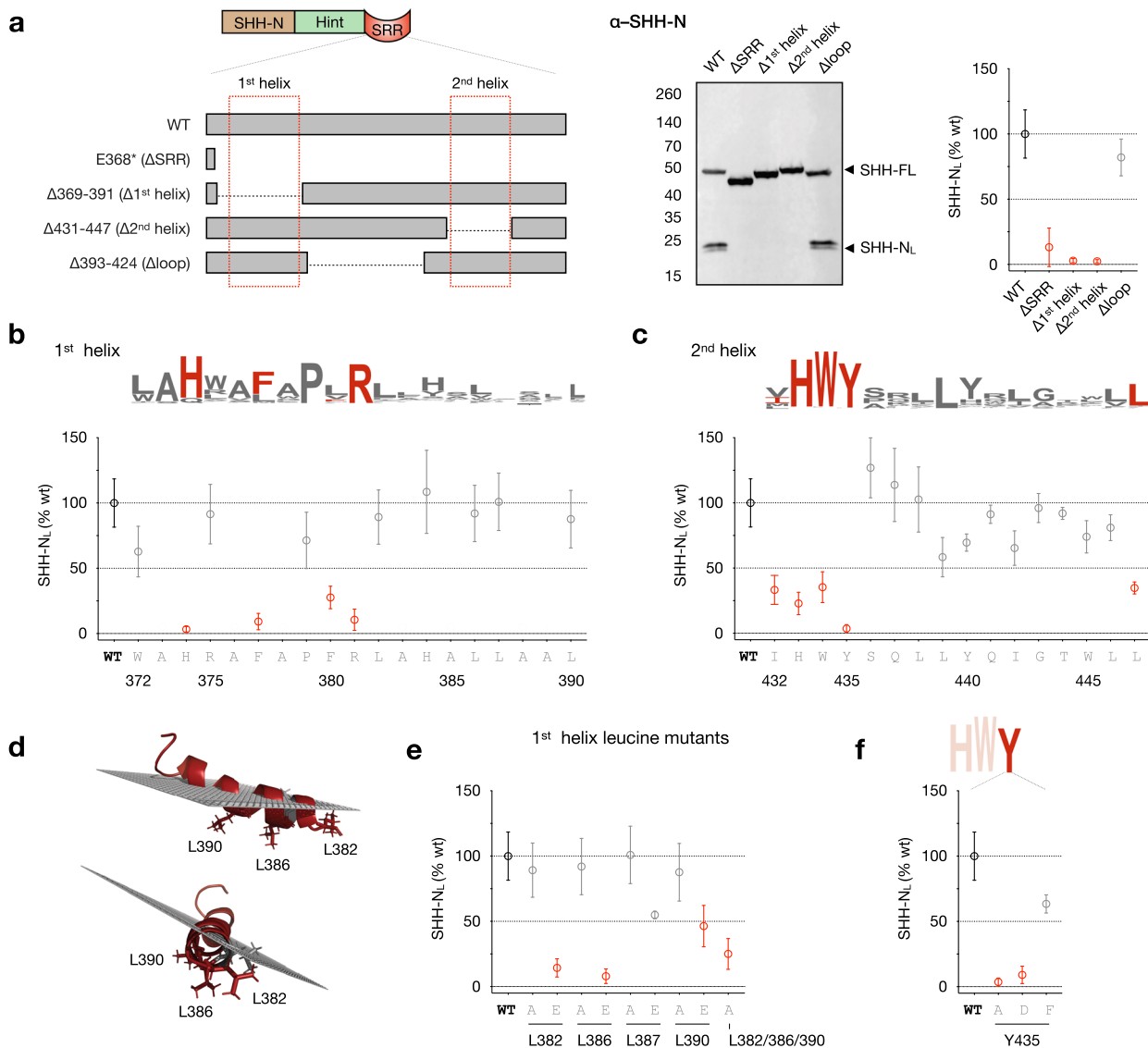

**Fig. 2 Conserved residues in the SRR helices enable cellular hSHH cholesterolysis. a** Left: Schematic of deletion mutants lacking individual SRR secondary structure elements. Center: Western blot analysis of cell-associated hSHH-N protein (hSHH-N$_L$) produced by SRR deletion mutants in overexpressing HEK293T cells. Right: Plot of relative hSHH-N$_L$ production versus wild-type protein for each mutant. **b** and **c** Alanine scanning of the 1st and 2nd helices, respectively, showing % hSHH-N$_L$ versus wild-type produced by each mutant. Above: sequence logos for the corresponding regions (from Fig. 1b). **d** Monte Carlo simulation of 1st helix residues W372-L390 in a phospholipid bilayer, showing leucine residues L382, L386, and L390 embedded in the membrane (ref. [37]). **e** Relative hSHH-N$_L$ produced by alanine and glutamate mutants of the three predicted membrane-embedded (L382, L386, L390A) residues and one surface (L387A) residue in the 1st helix of the SRR. **f** Relative hSHH-N$_L$ produced by alanine, aspartate, and phenylalanine mutants of Y435 in the 2nd helix of the SRR. For **a**, **b**, **c**, **e**, and **f**: The ratio of pixel intensity of hSHH-N$_L$ to hSHH-FL for each mutant was compared to the same ratio for wild-type protein and expressed as %WT. A biological replicate for wild-type protein was analyzed in each blot. Symbols represent the mean of $n =$ 3–10 biological replicates for each mutant ± s.d. Mutants that produced ≤50% hSHH-N$_L$ protein relative to wild-type protein are indicated in red.

production of cell-associated hSHH-N, a L382A/L386A/L390A triple mutant drastically reduced cell-associated hSHH-N, indicating that a stretch of alanine residues is insufficient to support the production of cholesteroylated hSHH-N in cells (Fig. 2e). Consistent with a model involving membrane association, replacement of either of the individual L382 or L386 residues with a negatively charged residue (glutamate) abolished production of cell-associated hSHH-N. An L390E mutation showed a distinct but attenuated reduction of cell-associated hSHH-N, whereas glutamate mutation of L387, which is perpendicular to the hydrophobic leucine surface, shows only slight reduction in cell-associated hSHH-N versus wild-type protein. This model also

suggests an electrostatic interaction between the basic residues H374 and R381 and phospholipid head groups, which may contribute to membrane association.[26]

The 2nd helix of the hSHH SRR contains a HWY motif that is largely conserved among species and Hh isoforms. While this motif has been demonstrated to regulate post-cleavage trafficking of hSHH-C,[39] a contribution to cholesterolysis has not been tested. Alanine mutation of each residue in this motif drastically reduced production of cell-associated hSHH-N (Fig. 2c). In particular, hSHH-N was undetectable in the lysates of cells expressing a Y435A mutant. We envisioned that this conserved tyrosine residue might participate in a hydrogen bond interaction

with the C3-OH of cholesterol. Surprisingly, mutation of this residue to an alternative hydrogen bond donor (aspartate) failed to rescue production of cell-associated hSHH-N. Interestingly, mutation to phenylalanine reduced total protein in lysate but restored the fraction of cell-associated hSHH-N relative to wild-type. The exclusive recovery of cell-associated hSHH-N by an aromatic residue suggests a potential mechanistic contribution of the Y435 π-system to cholesterolysis.[40] By contrast, the lower overall expression of Y435F suggests that the hydroxyl group is essential to biochemical and/or cellular protein stability.

The segment of the 2nd helix following the HWY motif consists of a number of polar residues interspersed with four conserved leucine residues (L438, L439, and L446, and L447). Surprisingly, while alanine replacement of the conserved L447 residue markedly decreased cell-associated hSHH-N, mutation of the less conserved residues L438, I442, and L446 to alanine had an attenuated effect (Fig. 2c). Alanine scanning of the remaining positions in the 2nd helix revealed that substitution of the polar residues S436 and T444 and the hydrogen bond donating residue Y440 did not substantially reduce production of hSHH-N in lysates. Likewise, mutation of glutamine residues Q437 and Q441 did not affect levels of cell-associated hSHH-N. Taken together, these studies provide a roadmap of the SRR residues involved in cholesterol transfer by Hh proteins in human cells.

**In vitro cholesterolysis requires a subset of SRR residues.** To distinguish between biochemical and cellular factors that contribute to cholesterolysis, we investigated the reactivity of SRR mutants toward cholesterol in vitro. To do so, we appended a Myc-DDK tag to the C-terminus of our hSHH construct (Supplementary Fig. 4a). Overexpression of tagged protein followed by mild lysis and affinity purification with agarose conjugated to an anti-FLAG antibody enabled us to isolate active unprocessed hSHH-FL free from N-terminal protein generated in cells. Consistent with previous observations, exposure of this protein to 0.5 mM cholesterol and 1 mM DTT resulted in production of cholesteroylated hSHH-N (hSHH-N$_C$), which could be distinguished from un-cholesteroylated hSHH-N by apparent molecular weight (Fig. 3a and Supplementary Figs. 4b and 5).[28,30,31,41,42]

For all mutants, exposure of the purified protein to 50 mM DTT effected production of hSHH-N, confirming formation of an active thioester. While a mutant lacking the catalytic cysteine residue (C198A) was unreactive, a construct lacking the entire SRR (E368*, ΔSRR) was susceptible to in vitro DTT cleavage but not cholesterolysis (Fig. 3b). Consistent with our cellular assays, neither the Δ1st helix nor the Δ2nd helix mutants were capable of undergoing cholesterolysis in vitro. Interestingly, a Δ2nd helix mutant underwent non-cholesteroylative cleavage in the presence of DTT and cholesterol, whereas a Δ1st helix mutant was unreactive under these conditions. This result suggests that an interaction between the 1st helix and cholesterol is sufficient to promote cleavage but not cholesterolysis. As anticipated, a Δloop mutant missing residues A393-P424 functioned equivalently to wild-type hSHH.

We next sought to query the in vitro reactivity of 1st helix SRR mutants that failed to produce cell-associated hSHH-N. All purified mutant proteins were capable of undergoing robust cleavage at high (50 mM) but not low concentrations (1 mM) of DTT (Supplementary Fig. 5). Consistent with our cellular assays, expression levels of the H374A and R381A mutants were lower than wild-type hSHH, and protein that was produced did not undergo cholesterolysis. While an F377A mutant expressed high quantities of protein, trace hSHH-N protein that was produced in the presence of 1 mM DTT and cholesterol likewise did not exhibit

a shift characteristic of cholesteroylated hSHH-N. Strikingly, while neither L382A/L386A/L390A, nor L386E mutants were capable of producing cell-associated hSHH-N, both exhibited a migration shift to indicate cholesterolysis in vitro. This observation suggests that the L382/L386/L390 residues act primarily to facilitate cellular aspects of cholesterolysis, potentially through an interaction involving this hydrophobic face within the 1st helix.

We then sought to characterize the biochemical contributions of the 2nd helix HWY motif to cholesterolysis in vitro. Purified H433A nor Y435A mutant proteins failed to produce cholesteroylated hSHH-N, consistent with their observed lack of this reactivity in cells. As in cells, mutation of the conserved Y435 residue to an alternative aromatic (Y435F) but not an aliphatic (Y435A) residue preserved biochemical cholesterolysis. By contrast, a W434A mutant, which was also incapable of producing cell-associated hSHH-N, showed restored reactivity toward cholesterolysis in vitro. The unique restoration of cholesterolysis by a W434A mutant indicates that this residue contributes predominantly to cellular aspects of hSHH cholesterolysis.

The observation that specific SRR mutants can undergo cholesteroylation in vitro but not in cells implies that hSHH-N cholesteroylation entails separable biochemical and cellular processes. Within our structural model, this analysis points to a cluster of conserved aromatic residues that is required for biochemical cholesterol adduction opposite to a hydrophobic face that is required for cellular cholesterolysis (Fig. 3c). Accordingly, this functional portrait provides a launching point for identification of related motifs in nature and a blueprint for redesign.

**The SRR contains a Golgi localization motif.** Due to its fleetingly low (nanomolar) concentrations in aqueous media, essentially all cellular cholesterol is sequestered in membranes. Moreover, specific lipid and protein compositions within a given membrane control the fraction of active cholesterol that is available to interact with other biomolecules.[43–46] While hSHH-N release has been investigated both functionally and biochemically in native hSHH-N secreting cells,[47,48] only a handful of studies have tracked the intracellular and extracellular trajectories of the Hint-SRR fragment.[31,36,49] Whether the SRR has inherent cellular targeting activity is unknown.

Because the hSHH Hint domain itself is soluble, we hypothesized that the SRR might localize the full-length hSHH protein to microdomains within the ER and or/other membranes for cholesterolysis. To ascertain preferences for subcellular localization inherent to the SRR, we created a fusion protein comprising N-terminal EGFP fused to SRR-encompassing residues A365-S462 of hSHH (Fig. 3d). To survey whether the SRR could bias the localization of the Hint domain, we also created EGFP-Hint (C198A), EGFP-Hint(C198A)-SRR fusions. When EGFP alone was expressed from a pCMV6-EGFP construct, it assumed a diffuse cellular distribution that included the nucleus. By contrast, when expressed as a fusion with the hSHH SRR, EGFP fluorescence was observed as discrete puncta clustered at the periphery of the nucleus, and nuclear fluorescence was eliminated. Time course studies revealed that this distribution was established immediately upon expression and at various levels of overexpression. Likewise, overexpression of an EGFP-Hint-SRR but not an EGFP-Hint construct was sufficient to establish punctate, extranuclear distribution. Finally, while deletion of the 1st and 2nd helices in an EGFP-SRR construct preserved extranuclear fluorescence, deletion of the 1st helix dramatically reduced puncta formation and enhanced cytoplasmic fluorescence, while deletion of the 2nd helix attenuated, but did not abolish, wild-type EGFP-SRR distribution. This observation is consistent with an important role for the 1st helix in subcellular trafficking, and reinforces a

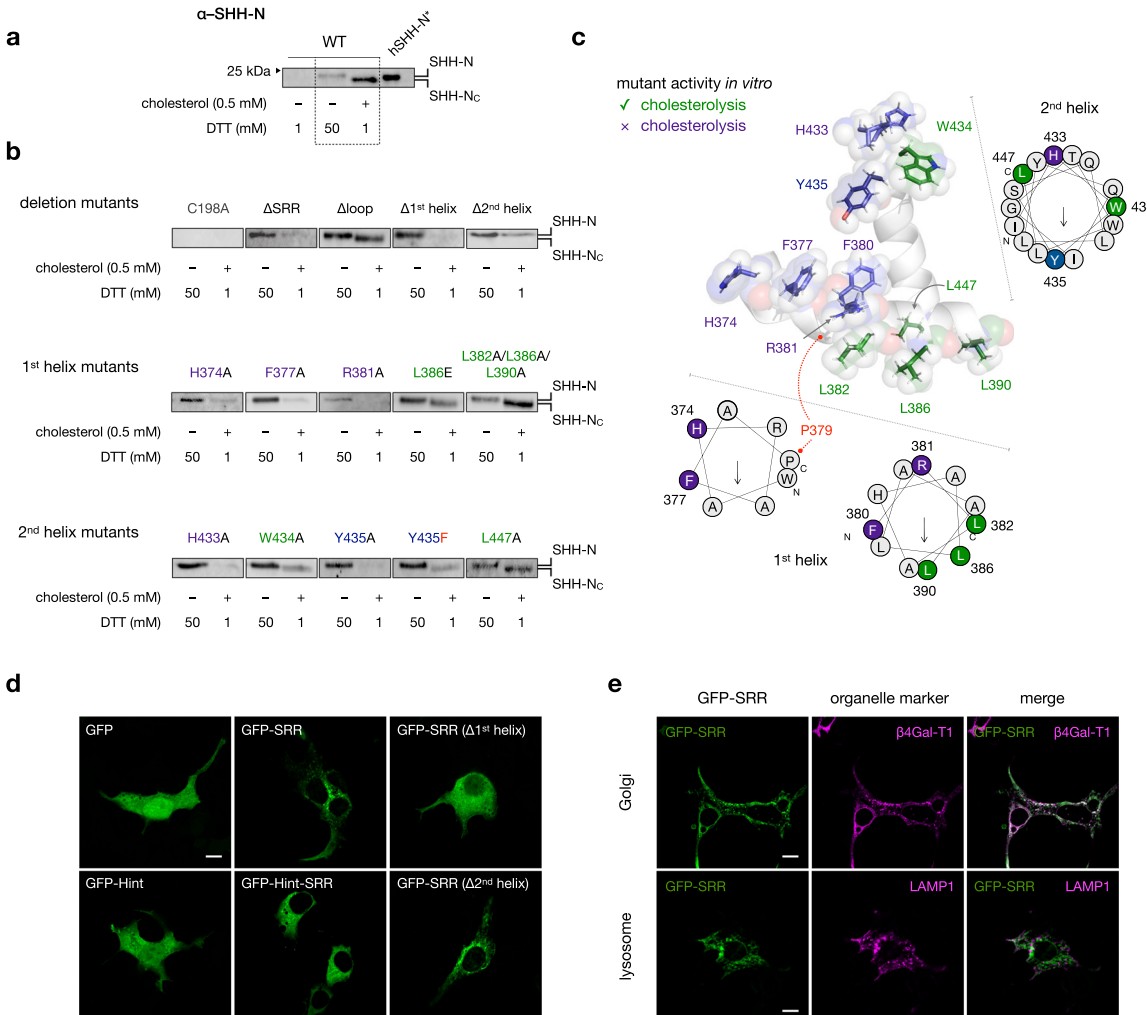

**Fig. 3 Specific SRR residues control biochemical cholesterolysis and cellular localization. a** Wild-type hSHH-FL isolated from HEK293T cells shows no cleavage in 1 mM DTT, non-cholesteroylative cleavage with 50 mM DTT, and cholesteroylative cleavage with 1 mM DTT + 0.5 mM cholesterol. Cholesteroylated hSHH-N (hSHH-N$_C$) shows a characteristic migration shift relative to non-cholesteroylated hSHH-N, as demonstrated by protein generated from a construct expressing hSHH-N only (hSHH-N*, residues 1-197). **b** Cleavage and/or cholesterolysis hSHH mutants by 50 mM DTT or 1 mM DTT + 0.5 mM cholesterol. For additional blots, see Supplementary Fig. 5. **c** SRR model showing residues required for cholesterol modification both in cells and in vitro (violet), in cells but not in vitro (green), and Y435 (blue), which is required both in cells and in vitro but can be functionally replaced by phenylalanine. Helical wheels show the facial positions of functional residues; the 1st helix is divided into coaxial segments before and after the P379 kink. **d** Confocal microscopy images of HEK293T cells expressing EGFP-SRR fusion proteins. **e** Images of EGFP-SRR coexpressed with mCherry fused to the Golgi-targeting sequence from β4-galactosyltransferase-1 (β4Gal-T1) or the lysosome-targeting sequence from LAMP1. Scale bar = 10 μM.

hypothesis that the 1st helix engages in functional membrane interactions during cellular cholesterolysis.

To determine which cellular compartments colocalize with EGFP-SRR, we co-expressed the EGFP-SRR construct with red fluorescent protein-labeled organelle markers. These studies revealed that SRR constructs colocalized extensively with mCherry markers bearing either β-galactosyltransferase or Golgin Golgi-targeting sequences (Fig. 3e and Supplementary Fig. 6). Importantly, the EGFP-SRR construct showed no indication of lysosomal localization to indicate misfolded or aberrantly processed protein. These studies indicate that the SRR alone imparts a distinct Golgi localization to fusion proteins, which may have important implications for Hh autoprocessing or secretion.

## Discussion
To elucidate the molecular features of the SRR that ligate cholesterol to Hh proteins, we have defined residues that are required

for cellular and biochemical Hh cholesterolysis. We used sequence conservation to identify common regions of secondary structure, which revealed a helix-loop-helix motif shared by all Hh proteins. Site-directed mutagenesis provided a blueprint of residues essential for Hh cholesterolysis in human cells and established minimal criteria for biochemical Hh cholesteroylation. This cellular and biochemical analysis was coupled to our identification of a new motif that induces Golgi localization in living cells.

Our data suggest possible mechanisms by which the SRR achieves this cholesterol-protein ligation in cells (Fig. 4). In one scenario, the SRR may tether the full-length Hh protein to cholesterol-rich membranes, enabling direct recruitment of a cholesterol nucleophile to the Hint active site. Alternatively, the SRR might itself extract cholesterol from the membrane and shuttle it to the reactive site within the Hint structure. Finally, the SRR could form a hydrophobic conduit with the Hint domain, and together this unit could orient cholesterol for nucleophilic attack.

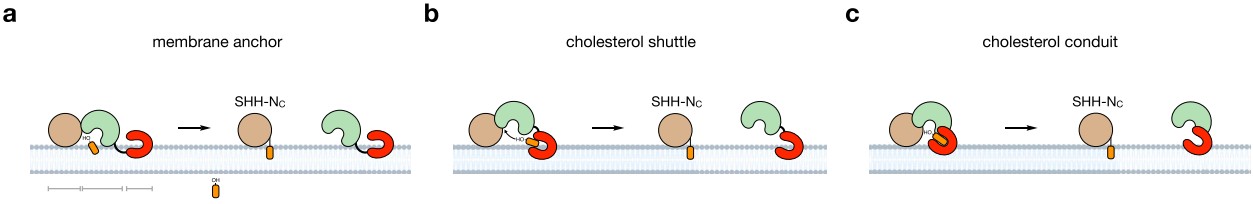

**Fig. 4 Models for SRR-promoted cholesterolysis of Hh proteins. a** The SRR localizes the Hint domain to a cholesterol-rich membrane interface, enabling direct access of cholesterol to the thioester. **b** The SRR extracts membrane cholesterol and delivers it to the thioester within the Hint domain. **c** The SRR interacts with the Hint domain to create a hydrophobic conduit, enabling cholesterol to access the thioester.

These studies can now provide a chemical basis for a number of SRR mutations that lead to developmental disease and can shed light on dysregulated Hh ligand activity in cancer (Supplementary Fig. 1). We are currently building upon this basis set of functional residues to re-engineer the SRR ligase activity for alternative substrates and proteins of interest. Future studies will investigate the contribution of Hint/membrane-SRR interactions to reaction efficiency to inform construct design. We anticipate that this work will lead to a general means to repurpose Nature's small molecule ligation technology as an invaluable tool in the chemical biology of living systems.

## Methods

**Antibodies**. Primary antibodies: Shh Antibody (E-1) (N-terminus), (Santa Cruz Biotechnology, sc-365112, 1:1000), Sonic Hedgehog/Shh C-Terminus Antibody (R&D Systems, AF445, 1:1000), Loading control: Karyopherin β1 (H-7) (Santa Cruz Biotechnology, sc-137016, 1:1000). Secondary antibodies: Donkey anti-Goat IgG (H + L) Cross-Adsorbed Secondary Antibody, Alexa Fluor 647 (Invitrogen, A-21447, 1:1000), Donkey anti-Mouse IgG (H + L) Cross-Adsorbed Secondary Antibody, Alexa Fluor 488 (Invitrogen, A-21202 1:1000).

**Constructs**. Myc-DDK-tagged human Sonic Hedgehog (hSHH, NP_000184.1) in a pCMV6 vector was obtained from Origene (RC222175). For analysis of proteins from cell lysates and media, a stop codon was introduced after the last residue of hSHH (S462) to obtain untagged hSHH. Untagged and Myc-DDK tagged hSHH constructs were used for site-directed mutagenesis to create the corresponding SRR mutants. Briefly, mutagenesis reactions were performed using overlapping or non-overlapping primers as required. Truncation mutants were generated by inserting stop codons via site-directed mutagenesis. SRR deletion mutants were generated by PCR extension of the parent hSHH sequence using phosphorylated reverse primers to exclude specified residues, then ligating the resulting PCR product using T4 ligase (NEB, M0318). To generate EGFP and EGFP-hSHH constructs used for microscopy experiments, an EGFP insert (Clontech, Addgene vdb2487) followed by the linker sequence (GGGS)₂ was cloned into the pCMV6 backbone before the corresponding hSHH fragment using Gibson assembly. All PCR reactions were performed using Phusion High Fidelity polymerase (NEB, M0530) in the presence of 10% DMSO. Parent construct was digested with DpnI (NEB, R0176). All primer sequences are available in Supplementary Methods.

**Protein sequence alignment and modeling**. Secondary structure prediction for human hSHH SRR (368–462) was performed using the online server JPred4 (http://www.compbio.dundee.ac.uk/jpred). Three dimensional models for the hSHH residues 363–462 were predicted by ab initio modeling using the Robetta online server (http://robetta.bakerlab.org). Helical wheel diagrams for the 1st helix (W372-A378 and P379-L390) and the 2nd helix (I432-L447) were generated using the Heliquest online server (http://heliquest.ipmc.cnrs.fr/). Hedgehog proteins were identified in the Uniprot repository using the query: name:hedgehog NOT name: skinny NOT name:acyltransferase NOT name:interacting NOT name:interfering NOT name:interference NOT name:hhat NOT name:like NOT name:receptor NOT name:cysteine NOT name:smoothened NOT name:sufu NOT name:tetratricopeptide NOT name:similar NOT name:putative NOT name:patched NOT name:domain NOT name:hint NOT name:pyruvate NOT name:amino NOT name:peptidase NOT name:quality NOT name:containing NOT name:partial length:[356 TO 518], and then by manual inspection. Protein sequences were aligned using the PRO-MALS3D algorithm (http://prodata.swmed.edu/promals3d) and visualized as a sequence logo generated by the WebLogo online server (http://weblogo.threeplusone.com/).

**Liposome preparation and CD spectroscopy**. Peptides for residues encompassing the 1st helix (368–391) and 2nd helix (431–449) of the hSHH SRR were obtained from Genscript Inc. at ≥90% purity. Crude soybean phospholipids

containing L-α-phosphatidylcholine (PC, Sigma-Aldrich, P5638) were used to prepare liposomes for CD analysis. Briefly, phospholipids in the presence or absence of cholesterol were dissolved in chloroform and dried to a thin layer on the sides of a glass vial using a rotary evaporator. Dried lipids were resuspended in CD buffer (specified below) and mixed to homogeneity using a vortex to yield a 10 mM PC ± 2 mM cholesterol stock solution. Unilamellar vesicles were prepared by disruption on ice using a tip sonicator, and liposomes were centrifuged at $4400 \times g$ for 10 min to remove undissolved lipids. Supernatant containing uni-lamellar vesicles was diluted to 0.5 mg/mL and analyzed by dynamic light scattering (Wyatt DynaPro PlateReader-II) to assess size distribution.

CD spectra were collected on 430 Series Aviv Biomedical Inc. spectrophotometer with a rotating sample chamber at 25 °C using a 0.1 cm path length quartz cuvette. CD scans were collected from 190–250 nm with a 1 nm bandwidth and 10 s averaging at each data point. Stock liposomes were diluted to a concentration of 0.5 mM PC ± 0.1 mM cholesterol in CD sample buffer. 1st helix peptide: A stock solution of peptide in aqueous 50 mM KF was diluted to 50–100 μM in liposomes ± cholesterol in 10 mM sodium borate (pH 9.6) and 50 mM KF. 2nd helix peptide: A solution of 50–100 μM peptide was prepared directly in liposomes ± cholesterol in 10 mM sodium phosphate (pH 7.4) and 50 mM KF. Signal for liposomes-only in the corresponding buffer was subtracted from the peptide signal for all measurements.

**Gel electrophoresis and Western blot analysis**. Samples were loaded on a 4–15% SDS-PAGE gel (Bio-Rad, 5678084) in Tris/Glycine/SDS running buffer (Bio-Rad, 1610772), and proteins were resolved at a current of 150 V for 1 h at room temperature. After electrophoresis, proteins were transferred to a 0.2 μM PVDF membrane (Bio-Rad, 1704157) using a Bio-Rad Trans-Blot Turbo Transfer System. Membranes were incubated with primary antibodies at a 1:1000 dilution in 2% milk in PBST (1× PBS and 0.1% Tween-20) for 6–12 h at 4 °C, washed three times for 5 min each with fresh PBST, then incubated with secondary antibodies at room temperature for 1–4 h. Fluorescence signal was collected on Bio-Rad Che-midoc MP Imaging System.

**Cell culture**. All hSHH proteins were expressed in adherent HEK-293T cells (ATCC, CRL-3216). Cells were cultured in high-glucose DMEM (Gibco, 11965118) containing 10% fetal bovine serum (FBS, Gibco, 26140079), 100 U/mL of penicillin-streptomycin (Gibco, 15140163), 1 mM sodium pyruvate, and 2 mM L-glutamine. Cells were seeded at an initial confluence of ~20%, passaged every 3–4 days upon reaching 85–90% confluence, and maintained in an atmosphere of 5% CO₂ and 95% humidity at 37 °C. Transfection mixtures were prepared in low serum media that contained 0.5% FBS instead of 10% FBS. Polyethylenimine (PEI, PolyScience, 23996-1) (1 mg/mL stock) was used as a transfection reagent.

**Analysis of hSHH protein from cells**. HEK-293T cells were seeded in a 12-well plate 24 h before transfection at a density of 200,000 cells/well. Upon reaching 80–90% confluence, cells were transfected with 0.75 μg hSHH plasmid DNA (untagged) and PEI at a ratio of 3:1 PEI:DNA (w/w), which were pre-incubated for 30 min in Opti-MEM (Invitrogen, 51985034) and diluted in low serum media. After media replacement, cells were incubated for 42–48 h. After rinsing with with 200 μL PBS, 125 μL ice-cold lysis buffer (50 mM Tris-HCl (pH 7.4), 250 mM NaCl, 1% IGEPAL CA-630, 1× complete EDTA-free Protease Inhibitor Cocktail (Roche, 11836170001) and 1 mM PMSF) was added to each well and the plate was incubated at 4 °C with rocking for 10–15 min. Cell material was collected by scraping and transferred to microcentrifuge tubes. Cells were then disrupted using a tip sonicator at 60% amplitude for 14 s (2 s ON, 1 s OFF). Cell lysate was heated to 98 °C for 5 min on a heating block, supplemented with SDS-PAGE sample buffer, boiled for additional 5 min, and analyzed or stored at −20 °C.

To detect secreted hSHH proteins, transfection media was collected before cell lysis and protein was precipitated with cold acetone and pelleted by centrifugation. The pellet was resuspended in 1× SDS-PAGE sample buffer, boiled for 5 min, and analyzed or stored at −20 °C.

**Purification of hSHH-FL**. Ten million HEK-293T cells were seeded in a 15 cm dish 24 h before transfection. Upon reaching 80–90% confluence, cells were transfected with 20 μg hSHH plasmid DNA (Myc-DDK tagged) and PEI at a ratio of 3:1 PEI:

DNA (w/w), which were pre-incubated for 30 min in Opti-MEM (Invitrogen, 51985034) and diluted in low serum media. Cells were incubated for 40–48 h, then dissociated with trypsin (TripLE, Thermo Fisher, 12605036), resuspended in culture media, and collected by centrifugation at $750 \times g$ for 5 min at room temperature. Cell pellets were flash frozen in liquid nitrogen and purified directly or stored at −80 °C.

All protein isolation procedures were performed on ice or at 4 °C. To isolate Myc-DDK tagged hSHH-FL, the cell pellet from each 15 cm plate was resuspended in 1 mL resuspension buffer (2× PBS (pH 7.4) + 5% glycerol + 1% Triton-X100 + 1 mM PMSF + 1× Roche EDTA free Complete Protease Inhibitor Cocktail). Cells were disrupted using a tip sonicator for 20 s (2 s ON and 1 s OFF), then lysate was solubilized by addition of Fos-Choline-12 (Anatrace, F308S) to final concentration of 0.5%. Lysate was incubated in at 4 °C for 30 min with gentle rocking to solubilize membrane-associated proteins, then clarified by centrifugation at $21,000 \times g$ for 10 min at 4 °C. Supernatant was carefully transferred to a new microcentrifuge tube and incubated with 100 µL pre-equilibrated Anti-Flag M2 Affinity Gel (Sigma, A2220) at 4 °C with gentle rocking for 2–3 h. Unbound proteins were removed by centrifugation and the beads were washed twice (4× resin volume, 15 min each) with wash buffer (2× PBS (pH 7.4) + 5% glycerol) containing 0.1% Fos-Choline-12. Bound proteins were eluted by incubating the resin for 2 h with 0.15 mg/mL FLAG-peptide in a final volume of 250 µL wash buffer containing 0.1% Fos-Choline12. Beads were removed from eluate containing hSHH-FL-Myc-DDK by centrifugation at $10,000 \times g$ and used immediately or flash frozen in liquid nitrogen and stored at −80 °C.

**Analysis of hSHH protein reactivity in vitro**. Isolated hSHH-FL-MycDDK protein was thawed on ice and divided into 30 µL aliquots in a PCR strip. For each experiment, 30 µL of the protein sample was boiled immediately after thawing to serve as an input control. For thiolysis, 1 M DTT was added to a final concentration of 50 mM. For cholesterolysis, 1 mM DTT was added from the 50 mM DTT stock, then 50 mM stock cholesterol in 2-propanol was added immediately with constant gentle vortexing, to avoid precipitation of cholesterol, to a final concentration of 0.5 mM. A 1 mM DTT-only reaction was performed similarly without cholesterol. The processing reactions were incubated at room temperature for 5 h with gentle rocking. Reactions were quenched by heating the samples to 98 °C for 5 min and adding 1× SDS-PAGE sample buffer, then samples were analyzed or stored at −20 °C.

**Fluorescence colocalization studies**. HEK-293T cells were seeded 24 h before transfection at 60,000 cells per well on sterile, acid cleared coverslips in a 24-well plate. Upon reaching 80–90% confluence, cells in each well were transfected with 0.2 µg GFP-hSHH SRR plasmid DNA and PEI at a ratio of 3:1 PEI:DNA (w/w), which was pre-incubated for 30 min in Opti-MEM (Invitrogen, 51985034) and diluted in low serum media. When present, the marker plasmids were included at 0.1 µg/well. After media replacement, cells were incubated for 6–8 h, then rinsed with 1× PBS and fixed with ice cold 4% paraformaldehyde for 10 min. Coverslips were rinsed three times for 5 min each with 1× PBS before a final rinse in ddH₂O, then mounted on clean glass slides in DAPI-free mounting media (Invitrogen, ProLong Diamond Antifade Mountant, P36965). Mounting media was cured in the dark for 8–12 h and coverslips were sealed with nail polish. Images were obtained on a Zeiss 700 confocal microscope using a ×63 oil immersion objective.

**Statistics and reproducibility**. For all hSHH mutant plasmids and EGFP fusion constructs, full hSHH insert sequences were verified by Sanger sequencing. For cellular cholesterolysis experiments, all graphical data points represent average Western Blot intensities from $n = 3–10$ biological replicates ± s.d. Band intensities were quantified using Bio-Rad Image Lab Software v6.0 and visually inspected for accuracy. For in vitro cholesterolysis experiments, electrophoretic mobility was standardized to calibrated protein markers and assessed using the Lane Profile feature of the Bio-Rad Image Lab Software 6.0. For fluorescence microscopy experiments, colocalization analysis was performed using the Coloc 2 plugin from ImageJ. Symbols represent the mean value of Pearson correlation coefficient ± s.d. from $n = 3–10$ cells from 3 separate experiments.

**Reporting summary**. Further information on research design is available in the Nature Research Reporting Summary linked to this article.

## Data availability

Western blot images for the data in Figs. 2 and 3 are provided in the Supplementary Information, along with all sequences of primers used to generate hSHH mutants. Western blot quantification for Fig. 2a–c, e, and f is available at https://doi.org/10.6084/m9.figshare.12133578.[36] Remaining data pertaining to this manuscript is available for corresponding author upon reasonable request.

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

## Acknowledgements
We thank Divya Kolli for preparation of the EGFP-SRR fusion construct, and Gracie Zhang and Colin Lee for performing Western blot control experiments. We are grateful to Prof. Peter Dervan and Prof. James K. Chen for valuable feedback. This work was supported by the Margaret E. Early Medical Research Trust.

## Author contributions
R.P. and A.E.O. designed experiments. R.P. performed CD spectroscopy, hSHH mutagenesis, cellular hSHH protein analysis, hSHH mutant protein isolation and in vitro cholesterolysis assays, and analyzed data. D.S.P. and E.V. performed hSHH mutagenesis, and E.V. performed Western blot experiments. A.E.O. performed EGFP-SRR mutagenesis, conducted microscopy experiments, and analyzed data. R.P. and A.E.O. wrote the manuscript.

## Competing interests
The authors declare no competing interests.
