## [Peer review file · Communications Biology]

Reviewers' comments:

Reviewer #1 (Remarks to the Author):

This manuscript by Ondrus and coworkers represents the first concerted effort to probe the structure/function of a cholesterol binding module of hedgehog proteins called the SRR. The SRR is a 50-100 amino acid segment found at the far C-terminal region of precursor forms of hedgehog (Hh) proteins. In precursor Hh, the SRR together with an adjacent intein-like domain, enable an unusual self-cleavage/auto-cholesteroylation activity. Cholesterol modification of Hh is fundamental to the far reaching roles of these proteins as morphogens and as oncogenic growth factors. Although a functional role for the SRR was assigned in general terms ~15 years ago, the atomic structure of this element has resisted experimental determination. As pointed out by the authors, there is also no unambiguous homolog in the PDB database, which suggests that the SRR, embedded within proteins of broad interest, represents an entirely new scaffold for binding one of nature's most important molecules. The SRR structure and its cholesterol-binding function have remained enigmatic because the subdomain appears to be intrinsically disordered, at least in part. In addition, the SRR is challenging to express in large quantities using *E. coli*, and lastly and least surprisingly, the SRR is lipophilic and aggregation prone.

Here the authors combine computational, biophy/biochemical and cellular techniques to analyze SRR function at the residue level for Sonic Hh protein, which is the most intensely studied of the three human Hh proteins. There are four major findings of the manuscript: 1) the computational model put forward as a hypothesis for the SRR structure; 2) experimental verification that two segments of the SRR are helix-forming in the presence of detergent; 3) that there is a substantial degree of permissiveness to SRR mutagenesis even at conserved positions; 4) that there is a subset of residues required for SRR function in cells but dispensable for SRR function in vitro.

Overall, the manuscript will be of significant interest to the field once the following concerns are addressed.

1. Page 1. The authors refer to the internal catalytic domain of Hh precursor as an "intein". Inteins have an accepted definition as self-splicing protein domains, which is distinct from the cholesterol ligase activity displayed by Hh. We suggest using "intein-like" or "HINT" for hedgehog/intein. Of these, HINT is most common. Not intein.
2. Page 1. Second paragraph, the term "Hh-C" is used for the first time. Please define what is referred to by Hh-C.
3. Page 3. The authors generate a 3D structure of Sonic Hh SRR. A PDB file should be supplied in the SI.
4. There appears to be inconsistency in SRR loop residue numbers. Described in the main body text (Page 3 line 6) as A391-G431. Elsewhere in paper (Figure 2A) the SRR loop residue is denoted as A393-P424. Please clarify.
5. Page 4 line 5-6. "The Δ loop construct that lacked a non-conserved 32 residue connector functioned identically to wild-type protein"

*What residues encompass this "32 residue connector"

*In Figure 3B, the WT construct and Δ loop construct appear to display distinct reactivity in vitro. The WT construct appears resistant to DTT reaction, while the Δ loop construct shows robust reaction with DTT. Is this difference significant?

6. Page 4. Alanine scanning mutagenesis is carried out for 28 residues in the SRR, according to Figure 2 B, C. We could not find the experimental data supporting the graphical summary in Figures 2 B, C. Reference is made to Supplemental Figure 3 but this figure has data for only 5 mutants. The missing data should be included in a revision.
7. Page 5 paragraph 4 line 3, refers to Fig S6 for reactivity of constructs with DTT or cholesterol, this does not appear to be the correct figure.
8. Page 5. In the third paragraph, the authors state that "Interestingly, a delta 2 helix mutant underwent non-cholesteroylative cleavage in the presence of DTT and cholesterol...." This is indeed a potentially interesting observation. The authors need to clarify how this mechanistic inference was reached.
9. Page 6. The authors propose a Golgi localization motif resides in the SRR. While potentially

exciting, this is the weakest element of the manuscript. We suggest that the authors consider removing the data or carrying out revised experiments to support their proposal. The EGFP fusion constructs used for these Golgi localization experiments are expressed from pCMV6. This vector does not appear to include an ER targeting leader sequence, which is the case with Human Hh proteins. Rather pCMV6 proteins are expressed in the cytosolic fraction as intracellular proteins. With this change in translation and trafficking of EGFP-SRR proteins compared to Hh proteins, we are left to wonder whether the apparent Golgi localization is meaningful. We suggest that the experiments should use a translation system that targets the secretory pathway, mimicking Hh protein translation.

10. Page 7 line 11 "... whereas no significant colocalization was observed with the nucleus, peroxisomes, mitochondria, actin, or the plasma membrane ...". Is this a qualitative assessment or was there statistical analysis. Please clarify.

11. Page 7 line 12 references a figure S7 which does not exist in the SI information.

12. The legend in Figure S4 lacks an explanation for Part A vs Part B

13. Figure S3 has description for a part C, but in the figure S3 there are parts A and B only.

14. Figure S6, is there a quantitative comparison of localization between Caveolae and Golgi? Caveolae localization could also be an interesting characteristic of hedgehog.

Reviewer #2 (Remarks to the Author):

This paper reports a sterol recognition region that ligate cholesterol to Hh proteins. The authors have used various approaches to characterize the molecular features of SRR that responsible for Hh cholesterolysis and have done a careful analysis. Overall the manuscript is well written, it provides new information on cholesterylation that are of interest in the field.

1. The authors have done a thorough mutagenesis study to specify residues within the conserved SRR region that are required for cholesterolysis in cells. However, residues in 1st helix of SRR showed different results in the biochemical assays from the cellular assays. It would be interesting to further study the membrane-binding properties of this helix region in vitro by using various amount of cholesterol.

2. The authors suggest a model of cholesterol transfer mechanism by SRR. Is it possible to monitor the cholesterolysis process in cells by using fluorescent sterols?

Reviewer #3 (Remarks to the Author):

In this manuscript, Purohit et al investigate the structural requirements for cholesteroliation in human hedgehog proteins. These proteins are noteworthy as they catalyze an unusual self-cleavage reaction during which the N-terminal fragment has a cholesterol moiety appended by the C-terminal fragment. This class of proteins represent the only known enzymes which function as cholesterol ligases. Mutations that alter this activity lead to congenital disorders, while loss-of-function is embryonically lethal – highlighting the importance of understanding hedgehog – cholesterol interactions. The authors focus on a fragment of human hedgehog found at the N-terminus of the SRR region which has been implicated in sterol interaction. Initial experiments identify a helix-loop helix motif which the authors confirm via CD using synthetic peptides. A series of deletions and alanine mutations of these region localize function to conserved residues within the predicted helices. The authors analyze function in a two-pronged approach. The first is a cell-based assay uses the localization of hedgehog post lysis as a proxy for cholesterol ligation. The second is an in vitro approach using affinity-purified hedgehog and a gel-based readout. Finally, the authors use a GFP-fused construct of the SRR to track the subcellular location of this C-terminal fragment. Using this construct, the authors demonstrate localization of GFP-SRR to the golgi.

The authors tackle a difficult, but significant question in a methodical manner. In the absence of high-resolution structural information, the use of alanine scanning and deletion constructs to assign function works nicely. Identification of the amino acids required for cholesterol ligation will

be useful for the field of hedgehog biology. In addition, data demonstrating subcellular localization will be useful for further experiments.

Comments:

#1 Further biochemical characterization of their mutants is warranted. Are certain variants more or less thermostable than others? Differential scanning fluorimetry may be appropriate due to (presumably) limited sample quantity.

#2 The authors briefly mention that the C-terminus of hedgehog is palmitoylated in the ER, yet they identify the C-term HRR fragment in the Golgi. Does the full-length intein-inactive variant (C198A) the authors have already generated also localize to the Golgi, or to a different subcellular compartment?

#3 It would be more convincing if the subcellular localization to the golgi could be demonstrated using and additional protein or two beyond GFP.

#4 The authors provide a supplemental figure detailing congenital disease and cancer associated mutations on the hedgehog sequence, but do not discuss these in depth. In particular, the cancer associated mutations do appear localized to the SRR region. The significance of the results may be bolstered by including functional data from a clinical variant.

Reviewers' comments: Reviewer #1 (Remarks to the Author):

This manuscript by Ondrus and coworkers represents the first concerted effort to probe the structure/function of a cholesterol binding module of hedgehog proteins called the SRR. The SRR is a 50-100 amino acid segment found at the far C-terminal region of precursor forms of hedgehog (Hh) proteins. In precursor Hh, the SRR together with an adjacent intein-like domain, enable an unusual self-cleavage/auto-cholesteroylation activity. Cholesterol modification of Hh is fundamental to the far reaching roles of these proteins as morphogens and as oncogenic growth factors. Although a functional role for the SRR was assigned in general terms ~15 years ago, the atomic structure of this element has resisted experimental determination. As pointed out by the authors, there is also no unambiguous homolog in the PDB database, which suggests that the SRR, embedded within proteins of broad interest, represents an entirely new scaffold for binding one of nature's most important molecules.

The SRR structure and its cholesterol-binding function have remained enigmatic because the subdomain appears to be intrinsically disordered, at least in part. In addition, the SRR is challenging to express in large quantities using *E. coli*, and lastly and least surprisingly, the SRR is lipophilic and aggregation prone.

Here the authors combine computational, biophy/biochemical and cellular techniques to analyze SRR function at the residue level for Sonic Hh protein, which is the most intensely studied of the three human Hh proteins. There are four major findings of the manuscript: 1) the computational model put forward as a hypothesis for the SRR structure; 2) experimental verification that two segments of the SRR are helix-forming in the presence of detergent; 3) that there is a substantial degree of permissiveness to SRR mutagenesis even at conserved positions; 4) that there is a subset of residues required for SRR function in cells but dispensable for SRR function in vitro.

Overall, the manuscript will be of significant interest to the field once the following concerns are addressed.

1. Page 1. The authors refer to the internal catalytic domain of Hh precursor as an "intein". Inteins have an accepted definition as self-splicing protein domains, which is distinct from the cholesterol ligase activity displayed by Hh. We suggest using "intein-like" or "HINT" for hedgehog/intein. Of these, HINT is most common. Not intein.

Thank you for pointing this out. We have replaced the term "intein" with "Hint" in the second paragraph of the introduction and the remainder of the text to distinguish between self-splicing intein domains and HINT domains. The "intein" label has been replaced with "Hint" in the block diagrams in Fig. 1A and Fig. 2A, and EGFP-intein/EGFP-intein-SRR have been replaced with EGFP-Hint/GFP-Hint-SRR in Fig. 3D. This terminology has also been corrected in Supplementary Figures, Supplementary Figure captions, and Methods. A review of these relationships by Dassa & Pietrokovski has been included in the references (ref. 10).

2. Page 1. Second paragraph, the term "Hh-C" is used for the first time. Please define what is referred to by Hh-C.

The term "Hh-C" has been removed from the text. The residues of the *D. melanogaster* Hh protein that are resolved in the crystal structure by Beachy et al. (residues 258-402, ref. 11) are specified as belonging to the Hint domain. "A crystal structure of the *D. melanogaster* Hint domain (residues 258-402) provided by Beachy and coworkers immediately revealed the basis for proteolytic activity of the protein: a splicing unit remarkably similar in tertiary structure to inteins found in bacteria and fungi."

3. Page 3. The authors generate a 3D structure of Sonic Hh SRR. A PDB file should be supplied in the SI.

A pdb file of the structure in Figure 1C is now provided in the SI.

4. There appears to be inconsistency in SRR loop residue numbers. Described in the main body text (Page 3 line 6) as A391-G431. Elsewhere in paper (Figure 2A) the SRR loop residue is denoted as A393-P424. Please clarify.

Thank you for this point. The "loop" region from our sequence conservation analysis is defined as the sequence between the 1st and 2nd helices (A391-G431). In our " Δ loop" construct we removed most, but not all, of this region to allow the 1st and 2nd helices to orient properly. We have clarified the definitions of "loop" and " Δ loop" in the text:

"Strikingly, a construct that lacked a non-conserved 32-residue connector (Δ loop, residues A393-P424) functioned identically to the wild-type protein."

5. Page 4 line 5-6. "The Δ loop construct that lacked a non-conserved 32 residue connector functioned identically to wild-type protein". *What residues encompass this "32 residue connector"?

We now specify the residues in the "32 residue connector" when mentioned in the text.

*In Figure 3B, the WT construct and Δ loop construct appear to display distinct reactivity in vitro. The WT construct appears resistant to DTT reaction, while the Δ loop construct shows robust reaction with DTT. Is this difference significant?

We cannot conclude from this data whether the observed difference is significant. Because cholesterolysis produced a reproducible, quantitative shift in the electrophoretic migration of hSHH-N, we chose to make our conclusions based on this parameter alone.

6. Page 4. Alanine scanning mutagenesis is carried out for 28 residues in the SRR, according to Figure 2 B, C. We could not find the experimental data supporting the graphical summary in Figures 2 B, C. Reference is made to Supplemental Figure 3 but this figure has data for only 5 mutants. The missing data should be included in a revision.

The requested data is now provided in Supplementary Fig. S3C.

7. Page 5 paragraph 4 line 3, refers to Fig S6 for reactivity of constructs with DTT or cholesterol, this does not appear to be the correct figure.

This has been corrected to Fig. S5 in the revised version. "All purified mutant proteins were capable of undergoing robust cleavage at high (50 mM) but not low concentrations (1 mM) of DTT (Fig. S5)."

8. Page 5. In the third paragraph, the authors state that "Interestingly, a delta 2 helix mutant underwent non-cholesterolylative cleavage in the presence of DTT and cholesterol...." This is indeed a potentially interesting observation. The authors need to clarify how this mechanistic inference was reached.

This is indeed a surprising result. Because non-cholesterolylated hSHH-N is largely secreted, we cannot detect it in our cell lysate cholesterololysis assays. The fact that we observed no production of cell-associated hSHH-N by a $\Delta 2^{\text{nd}}$ helix mutant (Fig. 2B) while we consistently observed production of non-cholesterolylated hSHH-N in our *in vitro* assays (Fig. 3A) might simply reflect this phenomenon. We hypothesize that cholesterol binding to the SRR induces a conformational change in the Hint-SRR; this might enhance the reactivity of the thioester toward non-cholesterolylative cleavage. We are currently extending our modeling and mutagenesis work to tackle this multi-part question.

9. Page 6. The authors propose a Golgi localization motif resides in the SRR. While potentially exciting, this is the weakest element of the manuscript. We suggest that the authors consider removing the data or carrying out revised experiments to support their proposal. The EGFP fusion constructs used for these Golgi localization experiments are expressed from pCMV6. This vector does not appear to include an ER targeting leader sequence, which is the case with Human Hh proteins. Rather pCMV6 proteins are expressed in the cytosolic fraction as intracellular proteins. With this change in translation and trafficking of EGFP-SRR proteins compared to Hh proteins, we are left to wonder whether the apparent Golgi localization is meaningful. We suggest that the experiments should use a translation system that targets the secretory pathway, mimicking Hh protein translation.

These points are well-taken. We agree that one cannot draw conclusions about possible roles of the SRR in hSHH secretory trafficking from these experiments. We used the pCMV6 vector (and omitted an ER-targeting sequence) to determine whether the SRR alone would bias the location of the Hint/EGFP construct. We have now quantitatively evaluated the overlap between EGFP-SRR and our $\beta 4$ -galactosyltransferase, Golgin, and LAMP1 markers with Pearson's colocalization coefficients (using the Coloc 2 plugin in ImageJ) with $n > 10$ cells for each marker. Per the Reviewer's suggestion, we have removed fluorescence images for markers in which data was collected in fewer than 3 separate experiments. In the revised version we show images only for coexpression of EGFP-SRR with $\beta 4$ -galactosyltransferase, Golgin, and LAMP 1, along with colocalization analysis results (Fig. S6B). We have modified the discussion to reflect these changes. To establish whether the SRR influences hSHH

trafficking through the secretory pathway we are developing a stable, inducible hSHH-expressing HEK293T Flp-In cell line to avoid artifacts from transient overexpression.

10. Page 7 line 11 "... whereas no significant colocalization was observed with the nucleus, peroxisomes, mitochondria, actin, or the plasma membrane ..." Is this a qualitative assessment or was there statistical analysis. Please clarify.

All images discussed in the revised manuscript have been re-processed and calibrated. Colocalization analysis for the β 4-galactosyltransferase, Golgin, and LAMP1-tagged markers for $n \geq 3$ images from three separate experiments are now provided in Fig. S6B.

11. Page 7 line 12 references a figure S7 which does not exist in the SI information.

This has been corrected to S6 in the revised version.

12. The legend in Figure S4 lacks an explanation for Part A vs Part B

This has been corrected in the revised caption for Fig. S4. "Figure S4. A C-terminal Myc-DDK tag enables isolation of hSHH-FL from overexpressing HEK293T cells and cholesterolysis *in vitro*. A. Western blot of HEK293T cells transfected with untagged and C-terminal Myc-DDK tagged hSHH-FL shows equivalent production of cell-associated hSHH-N (hSHH-N_L). B. Full blot showing *in vitro* cholesterolysis of wild-type hSHH protein."

13. Figure S3 has description for a part C, but in the figure S3 there are parts A and B only.

The caption for Fig. S3 has been corrected. The figure now includes Part C to describe Western Blot data for alanine scanning and point mutagenesis in Fig. 2B, C, E, and F.

14. Figure S6, is there a quantitative comparison of localization between Caveolae and Golgi? Caveolae localization could also be an interesting characteristic of hedgehog.

Our preliminary analysis suggests that overlap between the CAV1-mCherry marker and EGFP-SRR may be significant. For $n = 2$ experiments ($n = 5$ cells), Pearson's overlap coefficient for EGFP-SRR and mCherry fused to the targeting sequence of Caveolin 1 is 0.8 ± 0.1 . Per our decision to focus on images with analysis from 3 or more separate experiments, the image showing coexpressoin of EGFP-SRR and mCherry CAV1 has been removed from Fig. S6.

Reviewer #2 (Remarks to the Author):

This paper reports a sterol recognition region that ligate cholesterol to Hh proteins. The authors have used various approaches to characterize the molecular features of SRR that responsible for Hh cholesterolysis and have done a careful analysis. Overall the manuscript is well written, it provides new information on cholesterylation that are of interest in the field.

1. The authors have done a thorough mutagenesis study to specify residues within the conserved SRR region that are required for cholesterolysis in cells. However, residues in 1st helix of SRR showed different results in the biochemical assays from the cellular assays. It would be interesting to further study the membrane-binding properties of this helix region *in vitro* by using various amount of cholesterol.

Based on our hypothesis that the 1st helix interacts with the membrane, we anticipate that cholesterol does influence its membrane-binding properties. Unfortunately, we were unable to generate sufficient quantities of SRR protein for *in vitro* analysis. To address the important question of whether the secondary structure of the 1st or 2nd helices is cholesterol-dependent, we performed CD analysis in empty liposomes and liposomes loaded with cholesterol (Fig. 1E). This analysis, while limited, distinguished the 1st and 2nd helices of the SRR from cholesterol-binding amphipathic helices that require membrane cholesterol for helix formation (e.g. squalene monooxygenase). To address more subtle questions about the effect of cholesterol on SRR-membrane interactions, we are working to express soluble SRR constructs in *E. Coli* for *in vitro* studies.

2. The authors suggest a model of cholesterol transfer mechanism by SRR. Is it possible to monitor the cholesterolysis process in cells by using fluorescent sterols?

While Callahan et al. have demonstrated that fluorescent sterols (NBD, BODIPY) effect cleavage of a hSHH-N/Hh-C construct *in vitro* (refs. 18 and 22), these molecules are not cell-permeable. Several groups, including Tate *et al.*, have demonstrated that hSHH undergoes ligation to chemically modified sterols in mammalian cells, however, these methods are not suitable for live-cell imaging. Visualization of hSHH cholesterolylation in cells is complicated by membrane accumulation of the probes and high background fluorescence, We're working on chemical-genetic techniques to circumvent these issues, which could potentially be coupled to "turn-on" fluorescent probes that monitor hSHH cholesterolysis in live cells.

Reviewer #3 (Remarks to the Author):

In this manuscript, Purohit et al investigate the structural requirements for cholesterolylation in human hedgehog proteins. These proteins are noteworthy as they catalyze an unusual self-cleavage reaction during which the N-terminal fragment has a cholesterol moiety appended by the C-terminal fragment. This class of proteins represent the only known enzymes which function as cholesterol ligases. Mutations that alter this activity lead to congenital disorders, while loss-of-function is embryonically lethal – highlighting the importance of understanding hedgehog – cholesterol interactions. The authors focus on a fragment of human hedgehog found at the N-terminus of the SRR region which has been implicated in sterol interaction. Initial experiments identify a helix-loop helix motif which the authors confirm via CD using synthetic peptides. A series of deletions and alanine mutations of these region localize function to conserved residues within the predicted helices. The authors analyze function in a two-pronged approach. The first is a cell-based assay uses the localization of hedgehog post lysis as a proxy for cholesterol ligation. The second is an *in vitro* approach using affinity-purified hedgehog and a

gel-based readout. Finally, the authors use a GFP-fused construct of the SRR to track the subcellular location of this C-terminal fragment. Using this construct, the authors demonstrate localization of GFP-SRR to the golgi.

The authors tackle a difficult, but significant question in a methodical manner. In the absence of high-resolution structural information, the use of alanine scanning and deletion constructs to assign function works nicely. Identification of the amino acids required for cholesterol ligation will be useful for the field of hedgehog biology. In addition, data demonstrating subcellular localization will be useful for further experiments.

Comments:

#1 Further biochemical characterization of their mutants is warranted. Are certain variants more or less thermostable than others? Differential scanning fluorimetry may be appropriate due to (presumably) limited sample quantity.

While affinity-based enrichment of Myc-DDK-tagged proteins provided us with material for Western Blot analysis, these preparations were not sufficiently pure for differential scanning fluorimetry. Cellular cholesterolysis studies indicate that certain mutants are sensitive to thermal or cellular degradation. We now provide raw images of blots for Fig. 2C, 2D, 2F, and 2G in Fig. S3 to show differences in overall protein levels between mutants. We are moving forward with recombinant expression systems that can provide purified proteins for *in vitro* thermostability studies.

#2 The authors briefly mention that the C-terminus of hedgehog is palmitoylated in the ER, yet they identify the C-term HRR fragment in the Golgi. Does the full-length intein-inactive variant (C198A) the authors have already generated also localize to the Golgi, or to a different subcellular compartment?

Residue C24 of hSHH (in hSHH-N) is palmitoylated by Hedgehog acyltransferase after removal of the ER signal peptide, through mechanisms that do not involve the SRR. As observed by Salic (e.g. ref. 34) and others, a C198A mutant of hSHH does localize to the ER in a signal peptide-dependent manner. Because our constructs lack a signal peptide, ER localization is not anticipated. In general, proteins expressed from the pCMV6 vector show cytoplasmic distribution (e.g. EGFP alone, Fig. 3D, top left panel). This is true for the EGFP-intein construct (now referred to as "EGFP-Hint"), which does not show biased distribution (Fig. 3D, bottom left panel). By contrast, constructs that include the SRR accumulate in the Golgi (Fig. 3E and S6), suggesting that the SRR itself plays a dominant role in this localization. More detailed studies are required to determine the possible significance of this localization in hSHH cholesterolysis or trafficking.

#3 It would be more convincing if the subcellular localization to the golgi could be demonstrated using and additional protein or two beyond GFP.

To identify localization artifacts induced by EGFP fusion we included an EGFP-only vector control in our organelle marker coexpression experiments. We observed no overlap between EGFP alone and any of the organelle markers. We elected to use an EGFP as a reporter due to the availability of mCherry-tagged localization constructs (Addgene), which enabled co-expression and direct simultaneous monitoring. In future studies, we plan to analyze distribution of stably expressed hSHH-N in Flp-In cells (Invitrogen) through antibody colocalization.

#4 The authors provide a supplemental figure detailing congenital disease and cancer associated mutations on the hedgehog sequence, but do not discuss these in depth. In particular, the cancer associated mutations do appear localized to the SRR region. The significance of the results may be bolstered by including functional data from a clinical variant.

Figure S1 depicts only SRR mutations (updated 1/1/2020); mutations in the hSHH-N morphogen and Hint domain are not shown. Within the SRR, Roessler *et al.* (ref. S1) validate the role of 18 of these mutations in holoprosencephaly. The remainder of the mutations are derived from GWAS studies and their physiological relevance has not been explored. We are very much interested in pursuing the physiological, biochemical, and cellular consequences of these mutations, which is a significant part of our ongoing studies.

Main Figures

- Figure 1.** A. Replaced “intein” with “Hint”.
- Figure 2.** A. Replaced “intein” with “Hint”.
- Figure 3.** C. Changed the color-coding legend text.
D. Replaced original images with background-corrected images.
E. Replaced original images with background-corrected images used for colocalization analysis.
- Figure 4.** Replaced “intein” with “Hint”

Supplementary Figures

- Figure S1.** Replaced “intein” with “Hint”.
- Figure S2.** Replaced “intein” with “Hint”.
- Figure S3.** Added Western Blot images of cell-associated hSHH-N produced by each mutant, corresponding to Fig. 2B, C, E, and F.
- Figure S5.** Replaced “intein” with “Hint”. Added *in vitro* cholesterololysis data for a D243A mutant.
- Figure S6.** A. Removed all images collected in $n < 3$ separate experiments.
B. Added colocalization analysis for Fig. 3E and S6A.

REVIEWERS' COMMENTS:

Reviewer #1 (Remarks to the Author):

The authors have made a concerted effort to address the questions and concerns we raised. We expect that the work in its revised form will be of high interest and generate productive discussions about this mysterious transformation of Hh proteins.

We still harbor concerns about the biological significance of the results suggesting Golgi targeting of the SRR. However, we leave this to the considerations of the 2 other reviewers and editors; we are not cell biologists.

Brian Callahan

Reviewer #2 (Remarks to the Author):

The questions are well answered. The authors mention to study the effect of cholesterol on SRR-membrane interactions in vitro and monitor hSHH cholesterolysis in live cells in future work. A comment on membrane association would be helpful to the reader.

Reviewer #3 (Remarks to the Author):

The authors have thorough responses to initial reviews. The edited manuscript provided is significantly improved.

Inclusion of raw WB images in the supplemental is welcomed addition that helps to clarify differences in expression levels of their variants. A table quantifying expression of variants relative to WT could be helpful.

Reviewers' comments: Reviewer #1 (Remarks to the Author):

The authors have made a concerted effort to address the questions and concerns we raised. We expect that the work in its revised form will be of high interest and generate productive discussions about this mysterious transformation of Hh proteins. We still harbor concerns about the biological significance of the results suggesting Golgi targeting of the SRR. However, we leave this to the considerations of the 2 other reviewers and editors; we are not cell biologists.

We've assessed colocalization of our EGFP-SRR fusion construct containing residues 365-462 of human Sonic Hedgehog by coexpression with RFP fusion proteins containing two different Golgi targeting sequences (Fig. 3f and Supplementary Fig. 6). In our revised manuscript, we compare this data to the same analysis with an RFP fusion protein containing a lysosome targeting sequence. Evaluation of Pearson's coefficient for overlap between EGFP and RFP signals with the Coloc 2 plugin in the ImageJ software provides values of $78 \pm 4\%$ and $74 \pm 8\%$ for the Golgi-targeting markers β 4-galactosyltransferase-1, and Golgin, respectively, as opposed to $40 \pm 4\%$ for the lysosome targeting sequence from LAMP1, which we show in Supplementary Fig. 6b. In the text we emphasize that our data refers to these fusion proteins specifically; further studies on the localization of endogenous or overexpressed Hh proteins are needed to make conclusions about biological significance.

Reviewer #2 (Remarks to the Author):

The questions are well answered. The authors mention to study the effect of cholesterol on SRR-membrane interactions in vitro and monitor hSHH cholesterololysis in live cells in future work. A comment on membrane association would be helpful to the reader.

We've emphasized the importance of membrane association in our Discussion in the phrase "Future studies will investigate the contribution of Hint/membrane-SRR interactions...".

Reviewer #3 (Remarks to the Author):

The authors have thorough responses to initial reviews. The edited manuscript provided is significantly improved. Inclusion of raw WB images in the supplemental is welcomed addition that helps to clarify differences in expression levels of their variants. A table quantifying expression of variants relative to WT could be helpful.

Because we analyzed hSHH in lysates, it is not possible for us to determine whether a given SRR mutation affects protein stability, expression level, and/or secretion of the full-length protein. For example, full-length H270A mutant protein is efficiently released into the media (Supplementary Fig. 3a). Further studies are necessary to distinguish the factors that influence production of cell-associated hSHH-N for each mutant.